# Genome-Wide Selection Signal Analysis to Investigate Wide Genomic Heredity Divergence between Eurasian Wild Boar and Domestic Pig

**DOI:** 10.3390/ani13132158

**Published:** 2023-06-30

**Authors:** Xinming Wu, Haoyuan Zhang, Haoyuan Long, Dongjie Zhang, Xiuqin Yang, Di Liu, Guangxin E

**Affiliations:** 1College of Animal Science and Technology, Southwest University, Chongqing 400716, China; wxm155156@163.com (X.W.); swuzhanghy@163.com (H.Z.); asrimoom@gmail.com (H.L.); 2Institute of Animal Husbandry, Heilongjiang Academy of Agricultural Sciences, Harbin 150086, China; djzhang8109@163.com; 3College of Animal Science and Technology, Northeast Agricultural University, Harbin 150030, China; xiuqinyang@neau.edu.cn

**Keywords:** domestication, pigs, wild boar, selection signal analysis

## Abstract

**Simple Summary:**

In this study, genome-wide selective analysis of 371 publicly available genomes from worldwide domestic pigs and wild boars was conducted via FST and XPEHH methods to understand their genetic differences. Some common genes under selection in Eurasian domestic pigs suggest that domestication of pigs worldwide has undergone parallel selection in physiological phenotypes, such as reproduction, metabolism, growth, and development.

**Abstract:**

As important livestock species, pigs provide essential meat resources for humans, so understanding the genetic evolution behind their domestic history could help with the genetic improvement of domestic pigs. This study aimed to investigate the evolution of convergence and divergence under selection in European and Asian domestic pigs by using public genome-wide data. A total of 164 and 108 candidate genes (CDGs) were obtained from the Asian group (wild boar vs. domestic pig) and the European group (wild boar vs. domestic pig), respectively, by taking the top 5% of intersected windows of a pairwise fixation index (FST) and a cross population extended haplotype homozygosity test (XPEHH). GO and KEGG annotated results indicated that most CDGs were related to reproduction and immunity in the Asian group. Conversely, rich CDGs were enriched in muscle development and digestion in the European group. Eight CDGs were subjected to parallel selection of Eurasian domestic pigs from local wild boars during domestication. These CDGs were mainly involved in olfactory transduction, metabolic pathways, and progesterone-mediated oocyte maturation. Moreover, 36 and 18 haplotypes of *INPP5B* and *TRAK2* were identified in this study, respectively. In brief, this study did not only improve the understanding of the genetic evolution of domestication in pigs, but also provides valuable CDGs for future breeding and genetic improvement of pigs.

## 1. Introduction

The domestication of Asian native pigs has a history of over 8000 years [1], whereas European pigs are the result of hybridization between Near Eastern pigs (ancestors of Anatolian wild boars, which were domesticated 10,500 years ago) and European wild boars [2]. 

As is well known, animal domestication has led to significant changes in physiology and economic traits, including temperament, growth, and reproduction, in comparison to their wild ancestors [3,4,5]. For example, the *Yanbian* cattle in northern China use their rich coat and well-developed fat to withstand the cold [6], the high hemoglobin concentration in northeast African sheep may be related to high altitude adaptation, and the high expression of genes related to water and salt balance may also facilitate better adaptation to soil salinization [7]. The domestication of pigs has a close and complex connection with humans, and it is one of the symbols of human civilization. Currently, pigs not only provide necessary animal resources for humans, but also serve as important biomedical research models [8]. 

Previous studies have determined that Eurasian wild boars have two independent phylogenetic branches, and their chromosomal composition and structural differences are clearly manifested as two gradients of east and west [2,9]. The different genetic characteristics between European and Asian domestic pigs include fat content, fecundity, body type, and other aspects [10]. On the one hand, Asian pigs have rich phenotypes, such as large litter size, strong stress resistance, and excellent meat quality, due to different cultural habits and ecological geographical limitations [11,12,13]. On the other hand, European pigs are more prominent in terms of growth speed and lean meat percentage [14,15]. Animals from different geographical and social backgrounds undergo a series of consistent changes in the long process of domestication from wild environment adaptability to meeting human expectations. For example, the emotional dependence of animals on humans, which reduces the sensitivity of wild boars to their social environment and makes them easier to approach by humans, leading to adaptation with larger social gatherings [5,16]. In the present study, a total of 371 whole-genome resequencing datasets of domestic pigs and wild boars worldwide were utilized to analyze the genetic differences in domestication and breeding under different geographical backgrounds. The findings could help further understand the domestic history of Asian and European wild boars to pig formation, and promote the genetic improvement theory of local domestic pigs.

## 2. Materials and Methods

In this study, 371 public genomes of worldwide pigs were obtained, including 208 individuals from Asia and 163 individuals from Europe. All of the samples from each continent were classified into the wild group (WG) and the domestic animal group (CG) on the basis of their breeding and management history (Appendix A). The SNP genotype datasets of all individuals were downloaded from the public database Genome Variation Map (https://ngdc.cncb.ac.cn/gvm/ accessed on 3 January 2023), which was mapped on the basis of the pig reference genome (*Sscrofa* 10.2).

All 15,077,980 SNPs distributed on 18 autosomes were used for genome-wide selection (GWS) signal analysis, and the genetic parameter FST were calculated by vcftools, with a window length of 40 kb size and a step length of 20 kb [17]. Phase fixation was performed using beagle software (thread = 6), and then XPEHH was calculated by selscan software [18]. The results were taken as absolute values. The intersection genes of the top 5% of genes covering both parameters were defined as candidate genes (CDGs) under selection. ANNOVAR software was used for gene annotation in accordance with the annotation file of the ENSEMBL database (http://useast.ensembl.org accessed on 21 April 2023). The KOBAS database (http://kobas.cbi.pku.edu.cn/kobas3 accessed on 21 April 2023) was used for GO and KEGG function enrichment analyses on the candidate genes. Regarding diversity and haplotype distribution, PLINK contributed to the extraction of the SNP genotype within the coding region of CDGs, and the haplotype was reconstructed by DNAsp software [19]. The frequency distribution and phylogeny network of CDGs were built with network software (https://zzz.bwh.harvard.edu/plink/download accessed on 6 May 2023).

## 3. Results and Discussion 

A total of 164 (e.g., *ESR1*, *FYN*, and *ITPR3*) and 108 (e.g., *FOXO1*, *AKT3*, and *LYN*) CDGs were obtained from the Asian group (wild boar vs. domestic pig) and the European group (wild boar vs. domestic pig), by taking the top 5% intersected windows of FST (Asia > 0.0643332; Europe > 0.256492) and XPEHH (Asia > 0.720583; Europe > 0.732448) parameters, respectively (Figure 1). Only eight interacted genes (e.g., *TRAK2*, *MOS*, and *DPY19L1*) were found between both groups (Appendix A).

Within the Asian group, 83 of the 164 genes were enriched in 188 KEGG pathways (Appendix A), and 158 were enriched in 1372 GO terms (Appendix A). In the European group, 48 of the 108 genes were enriched in 157 KEGG pathways (Appendix A), and 106 were enriched in 1109 GO terms (Appendix A). Moreover, six of the eight genes from the Eurasian interaction were annotated in nine pathways, and seven were annotated in 77 GO terms. According to the classification of KEGG and GO annotations in the Asian group, the most significantly enriched pathways (corrected *p* < 0.05) belonged to metabolism, protein binding, and female gonadal development. In the European group, the most remarkably enriched pathways (corrected *p* < 0.05) included Fc gamma R-mediated phagocytosis, the FoxO signal transduction pathway, and fatty acid metabolism. A series of pathways, such as olfactory transduction, inositol phosphate metabolism, and progesterone-mediated oocyte maturation, were enriched by a large number of selected CDGs in both groups.

As is known, the majority of domestication centers of domestic pigs in the world are in Europe and Asia [20]. However, their diverse phenotypic and physiological differences are caused by natural selection and artificial selection within different domestication scenes. The present study was the first to analyze the differential pathways in Eurasia. Asian native pigs are widely considered as possessing high fecundity, excellent environmental adaptability, and rough feeding tolerance [21]. Here, a series of genes was identified to understand the genetic basis of these biological properties. For example, ESR1, an estrogen type 1 receptor, was universally confirmed to be related to pig fecundity; in particular, some of its genetic variations are significantly associated with pig litter size [13]. A previous study verified that the type I receptor of estrogen (ESR1) was regulated by miR-503 and ssc-miR-671-5p to influence fetal mortality in the endometrial tissue of Meishan pigs [22]. *MAPK14* was found to regulate the induction of porcine follicle stimulating hormone (FSH) and related proteins, and mediate the influence of maternal gut microbiota on mouse fetal growth [23,24]. Meanwhile, rich evidence has shown that *IDH1* and *FANCA* play a crucial role in the development and maintenance of ovarian function [25,26].

This study also found that a large number of immune-related genes (e.g., *FYN*, *NDUFA4*, and *PLIN1*) and pathways (e.g., complement and coagulation cascades and leukocyte trans-endothelial migration) are under selection in Asian domestic pigs. For example, protein tyrosine kinase fyn (FYN), an important immune kinase, was associated with porcine reproductive and respiratory syndrome viral infections [11]. *NDUFA4*, as a downstream gene of the cytochrome c oxidase regulator (MOCCI) during inflammation, plays an important role in improving host anti-infection and anti-inflammatory capacity [27]. 

The rapid genetic evolution of domestic pigs in metabolism and environmental adaptation could be related to strict artificial selection, comfortable living conditions, and medical security [27,28]. The relevant CDGs identified in the present study also reflect this finding. *SLC4A4* has been confirmed to be involved in the formation of bicarbonate transporters and affects pancreatic secretion; it also affects lactation and milk protein content in cattle [29,30]. Numerous studies have shown that crude fiber has an important effect on the intestinal health of animals [31]. Reports have also shown that *LYZ* upregulation could promote the proliferation and renewal of intestinal epithelial cells, resulting in the digestive system adapting better to a high fiber content diet [32]. The present study also identified CDGs that are associated with excellent meat quality in Chinese local pigs, such as the SNP variant at *CACNA2D1* intron, which was found to be significantly correlated with economic phenotypes, including leg gluteus muscle conductivity and eye muscle area (EMA) [33]. In addition, *DECR1* expression was found to be related to pork quality [12], and *COL3A1* was confirmed to be involved in the formation of collagen in the muscles [34].

Since the 18th century, Europe has started purposefully breeding commercial pig breeds on a large scale and achieved remarkable improvements in economic performance. In this study, a large number of genes (e.g., *FOXO1*, *AKT3*, *DAXX*, *PLPP2*, *AKT3*, and *CASP10*) under strong selection were found in European domestic pigs (including commercial pig breeds) compared with European wild boars, including those for muscle growth, nutrient metabolism, and emotional cognition. Studies also found a significant relationship between *FOXO1* and EMA, carcass lean meat percentage, and muscle fiber index [14]. Coincidentally, *DAXX* was identified as a CDG for the lumbar muscle area and lumbar muscle depth in Duroc pigs [15]. Some studies found that *AKT3* is one of the CDGs for the number of nipples and promotes milk fat production of mammals such as cattle [35,36]. *AKT3* is also involved in various biological functions, including inflammatory response, lipolysis metabolism, and muscle fiber characteristics [37,38,39]. Caspase (CASP10) is involved in the apoptotic cell death pathway, which could regulate the endometrial and placental functions during the estrous cycle of sows and is conducive to the attachment of embryos [40]. Meanwhile, a strong sense of taste could enable pigs to quickly establish new dietary habits [41]. Research has shown that *SCN9A* is involved in the formation of voltage-gated sodium ion channels in taste bud cells, playing an undeniable role in animals’ perception of taste [42,43].

A series of identified genes demonstrated the efforts of European commercial pig breeders to obtain high-feed returns. For example, several genes were enriched in the KEGG pathways for digestion and absorption. Lipid phosphate phosphatase 2 (LPP2), as a member of the LPP enzyme family, was related to the dephosphorylation of various lipid substrates and the synthesis of triglycerides [44,45]. *SLC7A9* participates in the formation of the amino acid transporter light subunit SLC7A9/b, which is related to protein digestion and absorption [46]. Dephosphorylated serine/threonine kinase 2 (SGK2) was proven to regulate gluconeogenesis in the liver [47]. In addition, the *ABCB11* gene encoding bile acid output pump was shown to be involved in the process of bile salt secretion [48]. The expression of ABCA1 (cholesterol transporter protein) was also clearly associated with lipid homeostasis [49]. Therefore, rich evidence suggests that genes with digestive and metabolic functions are constantly influenced by selection during the breeding process.

Finally, the convergent characteristics of Eurasian crossover genes were analyzed in this study. The olfactory transduction pathway is remarkable as a shared pathway in both groups. Some studies found that Duroc pigs contained a higher proportion of olfactory receptor domains than Tibetan wild boars, possibly indicating that domestic pigs rely more on odor recognition [37]. In fact, many studies confirmed that a strong olfactory ability helps pigs recognize their surroundings and compensate for visual impairments, which is beneficial for their survival in grazing and shedding environments [6]. 

Some studies also showed that a phosphatidylinositol phosphatase (INPP5B) deficiency affects spermatogenesis and maturation, leading to infertility [28]. Many studies confirmed that *MOS* played an important role in promoting the maturation of oocytes after the stagnation of the second meiosis metaphase, and *MOS* mutations could lead to the stagnation and fragmentation of early embryos and the formation of large polar bodies [50,51]. These findings may be one of the reasons for the low fertility of wild boars. Positive human–animal interactions greatly influence animal behavior, performance, and production, such as the influence of the emotional bond between humans and goats on their productive performance and the unique human–dog bond, as companion animals [52,53]. The sensitivity of pigs to the external environment and their emotional dependence on humans are also constantly being selected for during the long-term domestication process. *TRAK2* was confirmed to be involved in the formation of mitochondrial transport kinesin, and a previous study indicated that abnormal morphology of the DISC1-Miro-TRAK complex in neuronal dendrites resulted in the development of schizophrenia and depression [54]. Coincidentally, a genetic variant in *MFSD8* was identified to lead to neurodegenerative disorders and cognitive impairment [55]. 

Haplotype analysis of exons of *INPP5B* (Figure 2A) and *TRAK2* (Figure 2B) was conducted, and 10 and 6 SNPs from the coding region of both genes and 36 and 18 haplotypes were constructed, respectively. The results showed that haplotype polymorphism was obviously lower in European wild boars and domestic pigs than in Asia; meanwhile, the haplotype H_1 of *TRAK2* accounted for 258/261 and the haplotype H_1 of *INPP5B* accounted for 230/252 in the European domestic population, indicating that European domestic pigs may have been selected for by a uniform standard. A previous study found that a *PRKG1* haplotype from Chinese indigenous pigs was highly correlated with backfat thickness and waist depth of Duroc pigs [56]. This finding led to the consideration of the sequential relationship between artificial selection and genetic penetration in the formation of modern commercial pig breeds.

## 4. Conclusions

This study showed that CDGs such as *FOXO1*, *AKT3* and *SLC7A9* were selected for during the domestication of European domestic pigs, indicating their superiority in growth and development, feed conversion, and other traits. In Asian domestic pigs, CDGs such as *ESR1*, *MAPK14*, and *FYN* were selected for during the domestication process, indicating their outstanding performance in traits like fertility and resistance to stress. Meanwhile, the genes subjected to parallel selection in Eurasian domestic pigs were mainly *MOS*, *INPP5B*, and *TRAK2*, showing that the reproductive and nervous systems of domestic pigs were significantly different compared with wild boars.

## Figures and Tables

**Figure 1 animals-13-02158-f001:**
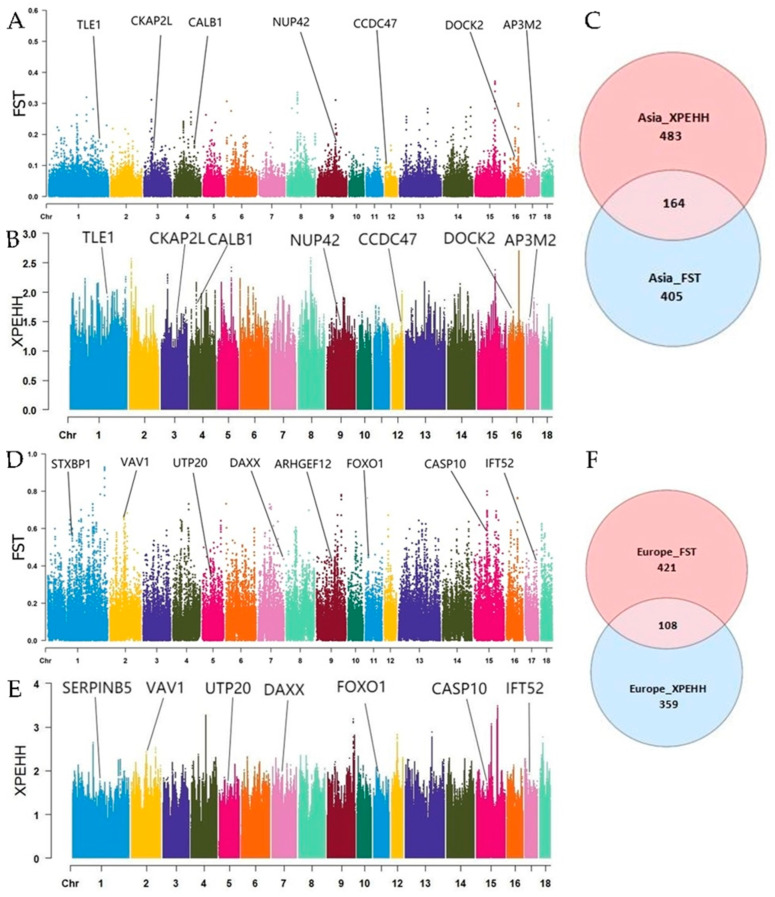
Genome-wide selective regions of Asian group (Wild boar vs. Domestic pig) and European group (Wild boar vs. Domestic pig). (**A**) Manhattan map of genomic FST in Asian group. (**B**) Manhattan map of genomic XPEHH in Asian group. (**C**) Venn diagram of interacted candidates in Asian group with FST and XPEHH parameters. (**D**) Manhattan map of genomic FST in Europe group. (**E**) Manhattan map of genomic XPEHH in Europe group. (**F**) Venn diagram of interacted candidates in Europe group with FST and XPEHH parameters.

**Figure 2 animals-13-02158-f002:**
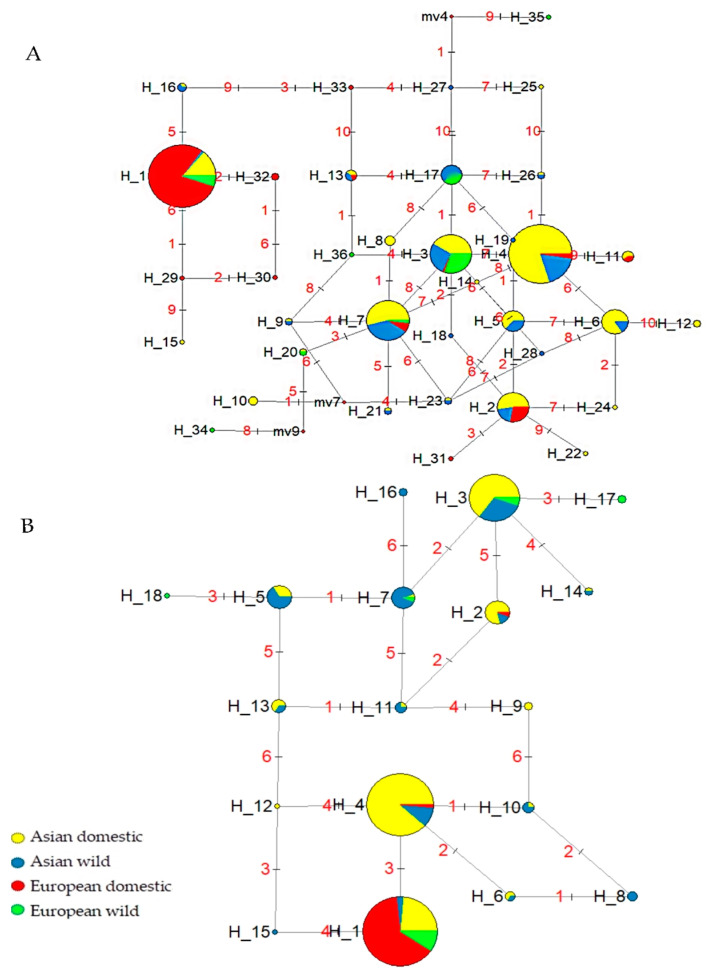
Phylogenetic network and frequency distribution of INPP5B and TRAK2 exon SNPs haplotype in worldwide pigs. (**A**) Distribution of 36 *INPP5B* haplotypes based on 10 exonic SNPs. (**B**) Distribution of 18 *TRAK2* haplotype based on 6 exonic SNPs.

## Data Availability

The SNP genotype datasets of all individuals were downloaded from the public database Genome Variation Map (https://ngdc.cncb.ac.cn/gvm/ accessed on 6 December 2022).

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
