# Peer review of "Genome-Wide Selection Signal Analysis to Investigate Wide Genomic Heredity Divergence between Eurasian Wild Boar and Domestic Pig"

_animals, 2023, doi:10.3390/ani13132158_

Round 1

Reviewer 2 Report

Dear Authors,

Wu et al. provide a very interesting investigation on the signature of genomic selection in domestic pigs (Asian and European), versus Eurasian wild boar under different geographical backgrounds. To do so, they used 371 publicly available whole-genome resequencing dataset obtained from domestic pigs, and wild boars, and applied FST and XPEHH. Using top 5% intersected windows of FST and XPEHH, the results indicate 164 and 108 candidate genes (CDGs) in Asian and European pigs, respectively. Although they found different selection pressures in Asian versus European domestic pigs, interestingly common genes related to physiological traits (reproduction, metabolism, growth, and development) went under parallel selection in both groups. The results presented in this manuscript has potential in contributing to the knowledge of the swine research community, and future breeding programs.

The manuscript is written clearly, the methodologies are adequately explained, and results are clearly presented (written and graphical), supporting the conclusion of the study. However, since the author are relating the selection pressure signature to differences in the geographical, and ecological differences experienced by Asian and European pigs, I highly recommend to add a paragraph in the discussion section describing this aspect in the other form of domestic animals (cattle, camel, sheep, etc.). The discussion is mainly focusing on the biological pathways on the candidate selected genes in pigs, and nothing on similar or different patterns in other livestock.

I very much enjoyed reading this paper, and I have some minor comments as below:

Line 13: rephrase to “… of 371 publicly available genomes from worldwide domestic pigs and … to understand the genetic differences ”.

Line 45: remove “Rich”, and instead provide more references to this conclusion.

Line 224, 226: in figure 2 inset, replace “Asian domestication” with “Asian domestic”, same for the European one.

Line 276: Table S3:

Line 278: Table S4:

Line 279: Table S5:

Line 280: Table S6:

Best Wishes,

Dear Author,

I would appreciate if the manuscript goes under English Editing prior to the publication. Particularly, some phrases are too long, and not very clear.

Best Wishes,
